# Expanding the Toolbox for Genetic Manipulation in *Pseudogymnoascus*: RNAi-Mediated Silencing and CRISPR/Cas9-Mediated Disruption of a Polyketide Synthase Gene Involved in Red Pigment Production in *P. verrucosus*

**DOI:** 10.3390/jof10020157

**Published:** 2024-02-16

**Authors:** Diego Palma, Vicente Oliva, Mariana Montanares, Carlos Gil-Durán, Dante Travisany, Renato Chávez, Inmaculada Vaca

**Affiliations:** 1Departamento de Química, Facultad de Ciencias, Universidad de Chile, Santiago 7800003, Chile; diegopalma.98@gmail.com (D.P.); bria.veog@gmail.com (V.O.); mmontanares@ug.uchile.cl (M.M.); 2Departamento de Biología, Facultad de Química y Biología, Universidad de Santiago de Chile, USACH, Santiago 9170022, Chile; cagild@gmail.com; 3Núcleo de Investigación en Data Science, Facultad de Ingeniería y Negocios, Universidad de Las Américas, Santiago 7500975, Chile; dtravisany@gmail.com

**Keywords:** CRISPR-Cas9, *Pseudogymnoascus*, red pigment, RNAi-mediated gene silencing

## Abstract

Fungi belonging to the genus *Pseudogymnoascus* have garnered increasing attention in recent years. One of the members of the genus, *P. destructans*, has been identified as the causal agent of a severe bat disease. Simultaneously, the knowledge of *Pseudogymnoascus* species has expanded, in parallel with the increased availability of genome sequences. Moreover, *Pseudogymnoascus* exhibits great potential as a producer of specialized metabolites, displaying a diverse array of biological activities. Despite these significant advancements, the genetic landscape of *Pseudogymnoascus* remains largely unexplored due to the scarcity of suitable molecular tools for genetic manipulation. In this study, we successfully implemented RNAi-mediated gene silencing and CRISPR/Cas9-mediated disruption in *Pseudogymnoascus*, using an Antarctic strain of *Pseudogymnoascus verrucosus* as a model. Both methods were applied to target *azpA*, a gene involved in red pigment biosynthesis. Silencing of the *azpA* gene to levels of 90% or higher eliminated red pigment production, resulting in transformants exhibiting a white phenotype. On the other hand, the CRISPR/Cas9 system led to a high percentage (73%) of transformants with a one-nucleotide insertion, thereby inactivating *azpA* and abolishing red pigment production, resulting in a white phenotype. The successful application of RNAi-mediated gene silencing and CRISPR/Cas9-mediated disruption represents a significant advancement in *Pseudogymnoascus* research, opening avenues for comprehensive functional genetic investigations within this underexplored fungal genus.

## 1. Introduction

The genus *Pseudogymnoascus* encompasses a diverse array of fungi that have aroused considerable interest in recent years. For instance, the species *Pseudogymnoascus destructans* is the etiological agent of white-nose syndrome (WNS), a pathogenic condition linked to substantial mortality rates observed in bat populations [1]. The emergence of WNS in 2009, coupled with advancements in the understanding of *Pseudogymnoascus* phylogeny [2], has stimulated global efforts to search for new members of the genus, particularly in soil environments, caves, and cold habitats. These efforts have yielded notable results. Among the total accepted *Pseudogymnoascus* species to date (amounting to 31 species according to MycoBank, Appendix A), 20 have been taxonomically characterized within the last five years [3,4,5,6,7,8].

From a biotechnological perspective, numerous studies have demonstrated that extracts derived from diverse strains of *Pseudogymnoascus* harbor bioactive metabolites with antibacterial, antifungal, trypanocidal, herbicidal, and antitumoral properties [9,10,11,12,13,14]. Furthermore, chemical analyses of *Pseudogymnoascus* strains have revealed that members of this fungal genus produce different classes of compounds, such as sesquiterpenes [15], macrolides [16], polyketides [17,18], tetrapeptides [19], and various nitrogenous compounds [20,21]. This underscores the potential of *Pseudogymnoascus* as a prolific source of specialized metabolites. It is worth noting that none of the specialized metabolites identified in *Pseudogymnoascus* have been linked to specific biosynthetic gene clusters (BGCs), leaving the molecular basis governing their biosynthesis in the fungus unknown.

To date, there are 26 publicly available genome sequences of *Pseudogymnoascus* strains in the GenBank database, encompassing known species such as *P. destructans*, *P. pannorum*, and *P. verrucosus*. This constitutes a valuable genetic reservoir for conducting functional and physiological studies in *Pseudogymnoascus*, particularly in characterizing BGCs responsible for producing specialized metabolites of interest. However, despite the availability of these genomic sequences, a significant hindrance to the molecular investigation of *Pseudogymnoascus* has been the lack of suitable tools for genetic manipulation. In recent years, important progress has been achieved in transforming different strains of *Pseudogymnoascus*, facilitated by the availability of suitable plasmids and the development of methodologies for protoplasts transformation, electroporation, and *Agrobacterium*-mediated transformation [22,23]. While these advancements have laid an important basis for the initial genetic manipulation of the fungus, the implementation of additional methods is necessary for the comprehensive functional study of its genes.

In fungi, two highly valuable and complementary techniques for functional genetic studies are RNAi-mediated gene silencing and CRISPR/Cas9-mediated disruption. RNAi-mediated gene silencing is grounded in the natural phenomenon known as RNA interference (RNAi), which is widely distributed across eukaryotes, including filamentous fungi [24]. In brief, RNAi is a post-transcriptional process where the presence of a double-stranded RNA molecule (dsRNA) is detected by the cell and subsequently processed into smaller RNA duplex molecules by a Dicer enzyme. One strand of the resultant RNA duplex integrates into an RNA-induced silencing complex (RISC), which is then guided to complementary sequences on the target mRNA. This leads to the degradation of the target mRNA through the endonuclease activity of RISC, resulting in the downregulation of gene expression [25]. RNAi-mediated gene silencing is straightforward as it only requires the plasmid-mediated expression of a dsRNA molecule from a small DNA sequence [24,25]. However, this technique does have some drawbacks, the main one being that the target gene remains intact within the genome. Consequently, even though fungi can achieve high silencing efficiencies [26], the technique does not completely suppress the expression of the target gene, resulting in a residual leaky expression.

In recent years, there have been significant advances in adapting CRISPR/Cas9 systems for application in filamentous fungi [27]. Like other organisms, the utilization of fungal CRISPR/Cas9 systems requires the expression of a Cas9 enzyme and a small chimeric protospacer adjacent motif (PAM)-dependent guide RNA (sgRNA). The sgRNA encompasses 20 nucleotides that are complementary to the target DNA region. When base pairing occurs, the Cas9 protein cleaves the sequence, creating a double-stranded break at this specific site [28,29]. This break can subsequently undergo repair through either homologous recombination or non-homologous end joining (NHEJ). In filamentous fungi, the prevailing repair mechanism is NHEJ, resulting in the introduction of small insertions or deletions (indels) in the target sequence. Consequently, this achieves functional disruption of the gene of interest.

The production of red pigments is a shared trait among various strains of *Pseudogymnoascus* [30,31]. Thus far, the precise chemical nature of these red pigments in *Pseudogymnoascus* has remained elusive. One interesting characteristic of this pigmentation is its conspicuous presence on agar plates, where the medium is typically stained with an intense reddish coloration, visible to the naked eye (see Results). We postulate that this pigmentation could serve as a visual indicator for evaluating targeting strategies directed toward the putative gene responsible for its biosynthesis. In this study, we successfully applied this proof-of-concept approach. Specifically, we sequenced the genome of a strain of *P. verrucosus* and identified a BGC that we hypothesize is responsible for the reddish coloration produced by the fungus. Through the implementation of RNAi-mediated silencing and CRISPR/Cas9-mediated disruption, we demonstrated that the *azpA* gene, which encodes a polyketide synthase within this BGC, is indeed responsible for producing the reddish pigment, thus validating our hypothesis. Additionally, at the molecular level, the success of the genetic manipulation was confirmed through qRT-PCR and sequencing. To the best of our knowledge, this is the first report describing the application of RNAi-mediated gene silencing and CRISPR/Cas9-mediated disruption in a fungus from the genus *Pseudogymnoascus*.

## 2. Materials and Methods

### 2.1. Strains, General Culture Conditions, and DNA Isolation

The fungal strain used in this study, *Pseudogymnoascus verrucosus* FAE27, was previously isolated from a sponge collected in Fildes Bay, King George Island, South Shetland Islands, Antarctica [10]. This strain and the transformants generated in this work were routinely cultivated at 15 °C in darkness on potato dextrose agar (PDA, BD Difco, Sparks, MD, USA) for a period of 15 days. For the inoculation of liquid cultures, conidia were harvested from these plates and subsequently transferred to 500 mL Erlenmeyer flasks containing 100 mL of CM medium (glucose 5 g/L, yeast extract 5 g/L, and malt extract 5 g/L). For the selection of transformants, PDA or solid CM media supplemented with hygromycin at a concentration of 60 μg/mL were used.

The routine cultivation of *Saccharomyces cerevisiae* S288 was carried out in YPDA medium (glucose 20 g/L, peptone 20 g/L, yeast extract 10 g/L, adenine 0.2 g/L, and agar 20 g/L).

DNA from *P. verrucosus* for PCR experiments was isolated from mycelium grown for one week in CM at 15 °C and 150 rpm. The mycelium was collected and washed with a 0.9% NaCl solution, and DNA was extracted following the procedure described by Mahuku [32].

### 2.2. Genome Sequencing and Identification of azp BGC

*Pseudogymnoascus verrucosus* FAE27 was cultured in CM medium at 15 °C and 150 rpm. After five days, the mycelium was harvested and washed with a 0.9% NaCl solution, and genomic DNA extraction was conducted using the QIAamp DNA Kit (Qiagen, Hilden, Germany) following the manufacturer’s instructions. This DNA was employed for PacBio sequencing, which was carried out by ADM Lifesequencing (Valencia, Spain). The genome has been assembled and is currently undergoing annotation (unpublished data).

Subsequently, the genome sequence of *P. verrucosus* FAE27 was submitted for analysis using the antiSMASH web server (fungal version 6.1.0) with default settings [33]. The deduced proteins from the genes constituting the BGC putatively responsible for reddish pigmentation, hereafter referred to as *azp* BGC, underwent BLASTP analyses to predict the putative function of each gene.

The nucleotide sequence of the *azp* BGC from *P. verrucosus* FAE27 described in this study has been deposited in the GenBank database under the accession number OR660048.

### 2.3. Construction of Plasmid pJLH-RNAi-azpA for RNA-Mediated Silencing of azpA Gene

For RNA-mediated silencing of the *azpA* gene in *P. verrucosus* FAE27, a suitable plasmid named pJLH-RNAi-azpA was constructed. A schematic diagram detailing the step-by-step construction process of this plasmid is depicted in Appendix A. The process was initiated with the assembly of a hygromycin resistance cassette. For this purpose, two DNA fragments were amplified via PCR. The first fragment, encompassing the P*gdh* promoter from *Aspergillus awamori*, was amplified using primers P1-KpnI and P2-pgdh-hph (Appendix A) with plasmid pJL43-RNAi [34] as the template. The second fragment, containing the *hph* gene conferring hygromycin resistance, along with the T*trpC* terminator from *Aspergillus nidulans*, was amplified using primers P3-pgdh-hph and P4-HindIII (Appendix A), and plasmid pAN7-1 [35] as the template. Both PCR fragments were subjected to in vivo recombination in *Saccharomyces cerevisiae* following the protocol outlined by Gietz et al. [36], with some modifications. In brief, 200 µL of competent yeast cells were transformed with a mixture containing 5 µL of each PCR product, 2 µg of linearized plasmid pRS426, 240 µL of 50% (*w*/*v*) PEG 3350, 36 µL of 1 M lithium acetate, and 50 µL of single-stranded salmon DNA (2 µg/mL) pre-heated. Following transformation, yeast cells were plated onto SC-Ura medium (glucose 20 g/L, Yeast Synthetic Drop-out Medium Supplements without uracil 0.2 g/L, Yeast Nitrogen Base without Amino Acids 6.7 g/L, and agar 20 g/L). Plasmids from the transformed *S. cerevisiae* colonies were extracted, propagated in *E. coli*, and subsequently subjected to sequencing to confirm the assembly. Once confirmed, the assembled hygromycin cassette was released by *Kpn*I and *Hind*III digestion and subsequently ligated into plasmid pJL43-RNAi [34], previously digested with the same enzymes, thus resulting in the obtainment of plasmid pJLH-RNAi.

Finally, a 413 bp fragment from the *azpA* gene from *P. verrucosus* FAE27 was amplified by PCR from genomic DNA using primers azpA-RNAi-Fw and azpA-RNAi-Rv (Appendix A). The fragment was digested with *Xba*I and subsequently ligated into the plasmid pJLH-RNAi, previously digested with the same enzyme, thus giving rise to the final plasmid pJLH-RNAi-azpA (Appendix A).

### 2.4. Construction of Plasmid pFC332-azpA for azpA Disruption by CRISPR-Cas9

The *azpA* sequence from *P. verrucosus* FAE27 was employed to select a target site for CRISPR-Cas9 disruption. To accomplish this, the software package sgRNACas9 v3.0.5 was utilized. This bioinformatic tool is designed to specifically identify Cas9 cleavage sites and predict potential off-target cleavage sites throughout a whole user-provided genome [37]. Following analysis with this software, a target sequence (5′CGTATTACACGGCCAGTAAC 3′) was selected based on its proximity to the 5′end of the gene and the absence of off-target sites. To verify the specificity of this target sequence, a comprehensive alignment was conducted between the chosen target sequence and the complete genome of *P. verrucosus* FAE27 using BLASTN. Subsequently, a 373 bp sgRNA expression cassette was custom-designed and synthesized by Integrated DNA Technologies (IDT, Coralville, IA, USA). This cassette was based on a previous design successfully used in *Penicillium roqueforti* and *Penicillium rubens* [38,39]. The cassette was delivered by IDT cloned into plasmid pUCIDT, and it encompasses the promoter and gene sequence of the proline-tRNA (tRNA^Pro1^) from *A. niger*, the sgRNA (which includes the target sequence), the terminator of tRNA^Pro1^ and a hammerhead ribozyme sequence positioned between the tRNA^Pro1^ gene sequence and the sgRNA. The cassette also included *Pac*I restriction sites at both ends for subsequent cloning (Appendix A).

The cassette was released from pUCIDT by *Pac*I and cloned into the vector pFC332 [40] previously digested with the same enzyme, resulting in the formation of plasmid pFC332-azpA. In addition to the sgRNA expression cassette, pFC332-azpA harbors a segment for Cas9 expression, the AMA region for autonomous replication of the plasmid, and the hygromycin resistance gene (Appendix A).

### 2.5. Transformation of P. verrucosus FAE27 and Transformants Selection

*Pseudogymnoascus verrucosus* FAE27 was transformed by polyethylene glycol (PEG)-mediated transformation of protoplasts, following the procedure detailed by Diaz et al. [23]. In each transformation experiment, 1 × 10^7^ protoplasts and 10 µg of plasmid were utilized in accordance with the recommendations outlined in that prior study [23]. Transformants resulting from the introduction of both pJLH-RNAi-azpA and pJLH-RNAi were subsequently transferred onto CM agar plates supplemented with 60 μg/mL hygromycin. Monosporic cultures of these transformants were obtained through the collection, dilution, and plating of conidia on selective media once again. To confirm the presence of the silencing plasmid within the transformants, a region of 1619 bp encompassing the fragment of the *azpA* gene and its flanking promoters was amplified by PCR using genomic DNA of the transformants, and primers RNAi-conf-fw and RNAi-conf-rv (Appendix A).

Homokaryotic strains derived from transformants resulting from the introduction of pFC332-azpA and pFC332 were obtained by seeding dilutions of transformant conidia onto PDA plates supplemented with 60 μg/mL hygromycin B. Subsequently, the strains were cultivated on PDA under non-selective conditions for seven to ten rounds to induce the loss of hygromycin resistance. Finally, to verify the disruption of the *azpA* gene in the transformants, sequencing analysis was conducted. Genomic DNA from both wild-type and transformant strains served as templates for amplifying the specific target region of *azpA* using primers Seq-Cas9-azpA-fw and Seq-Cas9-azpA-rv (Appendix A). The amplified regions were sequenced by Macrogen Inc. (Seoul, Republic of Korea).

### 2.6. RNA Extraction and qRT-PCR Experiments

For RNA extraction, *P. verrucosus* strains were grown on PDA plates at 15 °C for 14 days. Subsequently, mycelia were harvested using a sterile scalpel, frozen in liquid nitrogen, and then pulverized using a mortar. Fifty mg of pulverized mycelia were suspended in 1 mL of TriSure (Bioline, Memphis, TN, USA) and further disrupted using Tissueruptor (Qiagen, Hilden, Germany) to facilitate tissue breakdown. Following this step, total RNA extraction was carried out following established protocols [41]. Extracted RNA underwent treatment with RNase-free DNase I (New England Biolabs, Ipswich, MA, USA) to eliminate any potential DNA contamination. The total RNA concentration was determined using a NanoDrop ND-1000 spectrophotometer (Thermo Fischer Scientific, Braunschweig, Germany). Subsequently, one µg of total RNA was utilized for cDNA synthesis, employing the 5xAll-In-One 5X RT MasterMix (Applied Biological Materials, Richmond, BC, Canada) in accordance with the manufacturer’s instructions.

For gene expression analysis, qRT-PCR was performed with specific primers pairs: qRT-btub-Fw and qRT-btub-Rv for the β-tubulin gene, and azpA-Q-Fw and azpA-Q-Rv for *azpA* (refer to Appendix A). Reaction mixtures were prepared in 20 μL volumes, consisting of 10 μL of KAPA SYBR Fast qRT-PCR Master Mix 2x (Kapa Biosystems, Wilmington, MA, USA), 0.4 μL of each primer (at a concentration of 10 μM each), 0.4 μL de 50x ROX High/Low, 6.8 μL of water, and 2 μL of previously synthesized cDNA. qRT-PCR reactions were conducted using the StepOne Real-Time PCR System (Applied Biosystems, Waltham, MA, USA). The amplification conditions included an initial stage of 20 s at 95 °C, followed by 40 cycles of 3 s at 95 °C and 30 s at 50 °C. Three replicates were performed for the analysis. Data were analyzed using the comparative Ct (2^−ΔΔCt^) method and were normalized to β-tubulin gene expression in each sample.

### 2.7. Extraction of Red-Pigmented Metabolites and HPLC Analysis

For the extraction of red pigments, fungal strains were cultivated on PDA for 15 days at 15 °C. Subsequently, the mycelium was scraped off with a scalpel, and the remaining agar was cut into 1 cm pieces. These pieces underwent an overnight extraction with a mixture of methanol: ethyl acetate (1:1). Following extraction, the agar was removed, and the solvent was filtered through a 0.45 µm filter. The resulting filtrate was then evaporated to dryness using a rotary evaporator, followed by reconstitution in 500 µL of a methanol: water mixture (1:1) before being subjected to HPLC analysis. The HPLC system employed was a Waters HPLC system (Waters, Wexford, Ireland), consisting of a Waters 1525 Binary HPLC pump, a Waters 2996 Photodiode Array Detector, a Waters 2707 Autosampler, and a 4.6 × 250 mm (5 μm) SunFire C18 column. For the analysis, 50 µL of samples were injected into the HPLC system using a mobile phase comprising water (solvent A) and methanol (solvent B), both acidified with 0.01% trifluoroacetic acid. The elution gradient proceeded as follows: a linear increase from 10% solvent B to 100% solvent B over 20 min, followed by an isocratic phase for 8 min, and finally, a linear decrease from 100% solvent B to 10% solvent B over 2 min. The flow rate was set at 1 mL/min, and the column temperature was maintained at 35 °C.

## 3. Results

### 3.1. Identification and Characterization of azp BGC

The antiSMASH analysis of the *P. verrucosus* FAE27 genome revealed the presence of 29 BGCs. Among these, only one displayed significant similarity to known fungal BGCs associated with the biosynthesis of red-pigmented compounds. This specific BGC, denoted as *azp* BGC (Figure 1), shares structural resemblances with the ankaflavin and azanigerone BGCs found in *Monascus pilosus* and *Aspergillus niger*, respectively. Notably, both ankaflavin and azanigerone are classified as azaphilone-type pigments, suggesting that the *azp* BGC in *P. verrucosus* FAE27 may be responsible for the biosynthesis of compounds within this pigment family. It is worth noting that azaphilones can exhibit a color range from yellow to red based on their structural features. Therefore, the biosynthesis of azaphilone-type compounds by the *azp* BGC could explain the observed red pigmentation in the culture medium of *P. verrucosus* FAE27. Considering these findings, the genes within this BGC emerge as potential targets for the development of gene deletion and/or attenuation methodologies in *Pseudogymnoascus*. In order to identify the most suitable target gene, we conducted a bioinformatic analysis of the *azp* BGC.

The *azp* BGC spans a length of 49,400 bp and comprises 12 genes (Figure 1). To elucidate the potential functions of these genes, their deduced proteins were subjected to bioinformatic analyses using BLASTP. As depicted in Table 1, five genes encode proteins bearing similarity to those found in the ankaflavin BGC from *M. pilosus* [42], while two genes encode proteins with similarity to those in the azanigerone BGC from *A. niger* [43]. These findings provide further support that the *azp* BGC is responsible for the biosynthesis of an azaphilone compound. Noteworthy, two genes within the *azp* BGC encode enzymes of the polyketide synthase type (Table 1): *azpA* (Non-reducing PKS (NR-PKS); 8151 bp; 2698 aa) and *azpB* (Highly reducing PKS (HR-PKS); 7373 bp; 2315 aa). Currently, there are five known pathways for azaphilone biosynthesis, and all identified azaphilones derive directly from orcinaldehyde precursors synthesized by NR-PKS enzymes [44]. Consequently, we have identified the *azpA* gene as a potential target for knock-down and knock-out strategies.

### 3.2. RNAi-Mediated Silencing of azpA in P. verrucosus FAE27

Following transformation with plasmid pJLH-RNAi-azpA, 60 transformants were obtained, resulting in a transformation efficiency of 6 transformants/µg of plasmid. Among the 60 transformants, 8 (13%) displayed a completely white phenotype, characterized by the absence of pigmentation in both the colony and the agar. For subsequent analyses, we randomly selected three transformant strains with the white phenotype (azpA-pR1, azpA-pR17, and azpA-pR20), along with two transformant strains exhibiting wild-type phenotype (azpA-pR13 and azpA-pR15) (Figure 2A). All of them harbor the full silencing cassette in the genome (Figure 2B).

Quantitative analysis of *azpA* transcript levels demonstrated a remarkable 90% to 100% reduction in transformants displaying the white phenotype, in contrast to the wild-type strain (Figure 2C). Conversely, transformants exhibiting the red phenotype showed a more moderate decrease of 50–60% in transcript levels. Control strains that underwent transformation with plasmid pJLH-RNAi displayed no significant changes in gene transcript levels when compared to the wild-type strain (Figure 2C). This suggests that the plasmid pJLH-RNAi itself does not influence the observed phenotypic changes in the white transformants.

In order to confirm that *azpA* would be responsible for the red pigmentation in *P. verrucosus* FAE27, we conducted a comparative analysis of metabolite profiles between white and red-pigmented transformants. Initially, we examined the overall metabolic profiles of the strains. Extracts from various transformants obtained by RNAi-mediated silencing of *azpA* were subjected to HPLC analysis at 254 nm, revealing a general resemblance in metabolite composition, irrespective of coloration (Appendix A). However, when examined at 530 nm (the optimal wavelength for maximum red pigment absorbance), significant disparities became evident (Figure 2D). Under these conditions, extracts from red transformant strains displayed many peaks with maximal absorption at 530 nm within the retention time range of 10 to 18 min. In contrast, extracts obtained from the white transformants did not show any evidence of compounds with maximal absorption at 530 nm (Figure 2D). These findings strongly support the involvement of the *azpA* gene in the biosynthesis of metabolites responsible for the red staining of the culture medium.

### 3.3. Disruption of azpA Gene in P. verrucosus FAE27 by CRISPR-Cas9

To specifically target the *azpA* gene by CRISPR-Cas9, plasmid pFC332-azpA was generated, following the procedures outlined in the Materials and Methods section. Subsequently, we introduced this plasmid into *P. verrucosus* FAE27 through transformation. As a result, 21 transformants were obtained, yielding a transformation efficiency of 2.1 transformants/μg of plasmid. This efficiency is lower than the achieved with pJLH-RNAi-azpA, which is expected, considering that the size of pFC332-azpA exceeds more than twice that of pJLH-RNAi-azpA (15.6 kb vs. 7.3 kb).

Among the 21 transformants obtained, 16 (76%) displayed a completely white phenotype and showed no pigmentation in the agar. The remaining five transformants exhibited red pigmentation in the culture medium, resembling the wild-type strain. After obtaining monosporic cultures of all transformants, we inoculated them onto PDA plates without hygromycin, resulting in nine transformants (43%) losing hygromycin resistance. From this group, and for subsequent analysis, we selected all transformants with white phenotype (five transformants, namely azpA-pC54, azpA-pC57, azpA-pC61, azpA-pC63, and azpA-pC69) (Figure 3), and two transformants that retained a red pigmentation similar to the wild-type (azpA-pC16 and azpA-pC88, Figure 3). Additionally, we generated two control strains (pC1 and pC4, Figure 3) through transformation with pFC332, which were also included in the subsequent experiments.

As with transformants obtained by RNAi-mediated gene silencing, in transformants obtained through CRISPR-Cas9 methodology, we conducted a comparative HPLC analysis of metabolite profiles between white and red-pigmented strains. At 254 nm, the overall metabolic profiles displayed a general resemblance in composition regardless of coloration (Appendix A). However, when examined at 530 nm, extracts from red transformants exhibited multiple peaks within the retention time range of 10 to 18 min. In contrast, extracts obtained from the white transformants showed no evidence of compounds with maximal absorption at 530 nm (Figure 3). These results further support the involvement of the *azpA* gene in the biosynthesis of metabolites responsible for the red staining in *P. verrucosus* FAE27.

In order to conclusively establish that the *azpA* is responsible for the reddish coloration observed in *P. verrucosus*, we conducted sequencing of the target region of *azpA* in both the wild-type strain and selected transformants. As depicted in Figure 4, the nucleotide sequences of the red transformant strains, as well as control strains, remained identical to that of the wild-type strain. In contrast, all white transformant strains displayed an insertion in the target sequence, leading to a protein frameshift and the generation of a premature stop codon that effectively disrupts the *azpA* gene (Figure 4).

## 4. Discussion

Fungi from the genus *Pseudogymnoascus* exhibit interesting capabilities in their specialized metabolism [15,16,17,18,19,20,21]. To gain a deeper understanding of *Pseudogymnoascus’* metabolic potential, molecular tools enabling its genetic manipulation are essential. While several transformation systems for *Pseudogymnoascus* have been optimized [22,23], the options for gene manipulation in this important fungal genus have been limited due to the lack of suitable genetic tools. In this study, we contribute to filling this gap by implementing RNAi-mediated gene silencing and CRISPR-Cas9 techniques in the Antarctic strain *Pseudogymnoascus verrucosus* FAE27. To establish both methodologies, we take advantage of the ability of *P. verrucosus* FAE27 to secrete a red pigment into the culture medium. The red-pigmented phenotype is associated with the presence of a functional *azpA* gene. Consequently, the targeting of the *azpA* gene resulted in a white phenotype that can be easily distinguished from the wild-type phenotype by the naked eye. The availability of this visual reporter greatly facilitated the implementation of both techniques.

The analysis of the genome of *P. verrucosus* FAE27 revealed the presence of 29 BGCs. One of them, named *azp* BGC, exhibited similarity to BGCs associated with the production of two metabolites, ankaflavin and azanigerone, both belonging to the azaphilone family. Azaphilones constitute a structurally diverse family of fungal polyketides exhibiting absorbance spectra spanning from 330 to 530 nm. Consequently, they possess the capability to manifest yellow, orange, or red pigmentation depending on their specific chemical structure [45,46].

Azaphilones are synthesized by fungi spanning 61 genera, including *Aspergillus, Penicillium, Chaetomium, Talaromyces, Pestalotiopsis, Phomopsis, Emericella, Epicoccum, Monascus* and *Hypoxylon* [44,47]. Over time, chemical analyses of fungal azaphilones and the exploration of their metabolic pathways have supported the idea that these compounds are predominantly genus- or even species-specific [42,46,48,49], to the extent that azaphilones have been proposed as markers for taxonomic purposes in certain fungi [50,51,52]. The findings presented in this work suggest that *Pseudogymnoascus* is yet another fungal genus capable of azaphilone production. Further efforts, including the isolation of pure compounds and their spectroscopic characterization, will be necessary to elucidate the specific chemical structure of azaphilones originating from the genus *Pseudogymnoascus.* These endeavors, though, are beyond the scope of the present manuscript.

The core structure of azaphilones consists of a highly oxygenated pyrano-quinone bicyclic [53]. In the biosynthesis of this distinctive functional group, NR-PKS is involved [44]. Consequently, we selected the *azpA* gene as our target, as it was the sole gene encoding NR-PKS within the *azp* BGC. Disrupting the *azpA* gene through CRISPR-Cas9 or achieving substantial RNAi-mediated silencing (at least 90% reduction) led to the generation of white transformants that exhibited no staining in the culture medium. This strongly supports the notion that the *azpA* gene is essential to produce red compounds in *P. verrucosus*. In relation to the other genes within the *azp* BGC, our bioinformatic analysis provides further support indicating the involvement of this BGC in the biosynthesis of compounds belonging to the azaphilone family.

To date, five main biosynthetic pathways for azaphilones have been elucidated [44,47]. Functional analysis of genes in these pathways has been conducted employing various molecular techniques such as homologous recombination-based deletions, overexpression, heterologous expression, and in vitro reconstitution of selected reactions [44,47,54]. It is noteworthy that, as of now, RNA-mediated gene silencing has not been applied to investigate azaphilones BGCs. In the case of CRISPR-Cas9, this methodology was employed in *M. ruber* to inactivate two negative regulatory genes within the azaphilone BGC, resulting in an enhanced pigment production capacity [55]. To the best of our knowledge, no other examples of CRISPR/Cas9 application for studying azaphilone BGCs have been described.

RNAi-mediated gene silencing is a valuable technique for studying the functionality of fungal genes. Consequently, there has been considerable interest in implementing this technique across various fungal species [26,56]. In several studies using this technique, genes within melanin biosynthesis clusters have been targeted [57,58,59,60,61], following the same rationale as applied in this investigation. Strains deficient in melanin exhibit an albino phenotype in contrast to the corresponding wild-type fungus. Thus, the successful silencing of genes within melanin BGCs results in an easily distinguishable change in phenotype. It is noteworthy that in these studies, RNAi-silenced transformants with phenotypes closely resembling the wild-type were obtained [57,58,59,60,61]. Our findings concerning RNAi-mediated silencing of the *azpA* gene in *P. verrucosus* are analogous to these results. In our case, following the RNAi-mediated silencing experiment, we observed that 13% of the transformants displayed visibly distinct phenotypes compared to the wild-type strain, as evidenced by a colorless culture medium. This correlated with a reduction of over 90% in *azpA* transcripts. Regarding the transformants exhibiting a reddish phenotype identical to the *P. verrucosus* wild type (constituting 83% of transformants), they displayed only around 60% reduction in *azpA* transcripts. Therefore, a significant decrease in *azpA* transcripts is necessary to achieve distinctive phenotypes between these transformants and the wild-type strain of *P. verrucosus*.

The disruption of the *azpA* gene via CRISPR-Cas9 yielded more consistent results compared to RNAi-mediated gene silencing. The disruption of *azpA* effectively resulted in the gene´s inactivation, leading to transformants exhibiting phenotypes characterized by pronounced and unequivocal alterations, in contrast to those observed with RNAi-mediated gene silencing. Furthermore, the efficiency of *azpA* gene disruption through CRISPR-Cas9 was notably higher, with 76% of the transformants displaying the white phenotype, in contrast to the 13% observed with RNAi-mediated gene silencing. These results indicate that CRISPR-Cas9 is more effective in functionally disrupting *azpA* than RNAi-mediated silencing. This is expected, considering the distinct mechanisms of these methods. As mentioned, CRISPR-Cas9 induces a double-strand break, repairable by cellular mechanisms, leading to *azpA* inactivation. In contrast, RNAi-mediated silencing degrades mRNA but keeps the *azpA* gene intact in the genome. Consequently, with CRISPR-Cas9, *azpA* is completely inactivated, whereas with RNAi-mediated silencing, despite achieving high silencing efficiencies, complete suppression of *azpA* expression is not always attained. This partial suppression results in a residual leaky expression, which, in some cases, is sufficient to manifest the red phenotype produced by azaphilone pigments, known for their intense coloration [62,63]. Thus, in mutants that did not achieve substantial RNAi-mediated silencing of *azpA* (at least a 90% reduction in transcripts), the leaky expression induced fungal coloration, as shown in Figure 2. This contrasts with CRISPR-Cas9 methodology, where *azpA* was entirely inactivated, preventing any manifestation of the red phenotype.

The examination of CRISPR-Cas9-disrupted transformants revealed a consistent type of mutation (a T insertion) in the target sequence among those displaying the white phenotype. This represents a highly reproducible event. Previous investigations have identified two dominant categories of mutations occurring in the context of DNA double-strand break repair: NHEJ-dependent single-base insertion (entailing duplication of a nucleotide in position -4 from PAM) and MH-mediated deletion [64]. In our case, the single-base insertion corresponds to a thymine at the -4 position relative to the PAM site in the designed sgRNA, thus corresponding to the NHEJ-dependent single-base insertion mechanism.

In this investigation, we implemented RNAi-mediated gene silencing and CRISPR-Cas9 gene disruption methodologies for the analysis of the *azpA* gene in *P. verrucosus*. Beyond establishing this proof-of-concept, the methodologies outlined herein expand the repertoire for conducting functional analyses of diverse genes within the wide range of *Pseudogymnoascus* species. The choice between these two techniques will depend on the specific research objectives of the investigators. As suggestions, we envisaged several scenarios where one technique might be more convenient or suitable than the other. CRISPR-Cas9 is well-suited when precise and permanent genetic modifications are required; however, its application necessitates access to the sequenced genome of the specific *Pseudogymnoascus* species under investigation. Consequently, the methodology described in this study could be adapted for functional gene analyses in *Pseudogymnoascus* strains with available genomes, including species such as *P. destructans*, *P. pannorum*, and *P. verrucosus* [65,66]. Of particular interest would be the functional study of genes suspected to encode virulence factors in *P. destructans* [67], the etiological agent of WNS in bats. In contrast, RNAi-mediated gene silencing is advantageous in scenarios where only a partial coding region of the target gene is known or in cases where comprehensive genome sequences are lacking, a current circumstance for the majority of *Pseudogymnoascus* species. Moreover, given that RNAi-mediated gene silencing induces gene knockdown without gene elimination, it emerges as a valuable strategy for elucidating the function of genes essential or lethal in *Pseudogymnoascus*, as demonstrated in other filamentous fungi [68,69]. Furthermore, the straightforward application of RNAi-mediated gene silencing positions it as a good choice for high-throughput functional genomics screens, enabling the systematic evaluation of gene silencing effects on diverse cellular processes. In this context, the genomic analysis of *Pseudogymnoascus* underscores the presence of numerous biosynthetic gene clusters [16], thereby rendering RNAi-mediated gene silencing useful for the simultaneous experimental analysis of these genes, aiming to establish associations with the production of their respective metabolites.

In summary, this study represents the first implementation of RNAi-mediated gene silencing and CRISPR-Cas9 gene disruption methodologies in *Pseudogymnoascus*. The availability of these two distinct gene manipulation techniques expands the toolkit for conducting functional analyses of genes in this understudied fungal genus.

## Figures and Tables

**Figure 1 jof-10-00157-f001:**
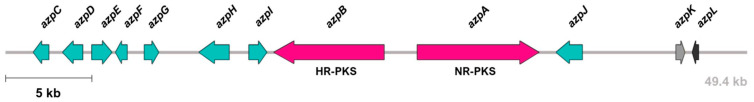
Schematic representation of the putative azaphilone biosynthesis gene cluster (*azp* BGC) in *Pseudogymnoascus verrucosus* FAE27. Arrows indicate the genes and their respective transcriptional orientation. Refer to Table 1 for the predicted functions of each gene.

**Figure 2 jof-10-00157-f002:**
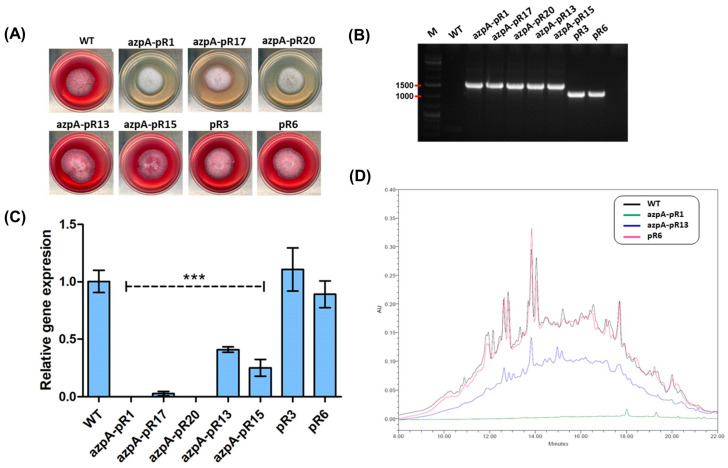
RNA-mediated silencing of *azpA* in *P. verrucosus* FAE27. (**A**) Phenotypic variations observed in selected *azpA* transformants following the introduction of pJLH-RNAi-azpA. *P. verrucosus* FAE27 (WT) exhibits its characteristic red pigmentation, while transformants azpA-pR1, azpA-pR17, and azpA-pR20 display a white phenotype. Transformants azpA-pR13 and azpA-pR15 retain the wild-type appearance. Additionally, two control transformants (pR3 and pR6), carrying the empty pJLH-RNAi plasmid lacking the RNAi silencing cassette, were included. These control transformants display the same phenotype as the wild-type fungus. (**B**) PCR assay confirming the integration of the full silencing cassette in the transformants. The presence of a 1619 bp amplicon confirms the insertion of the cassette in all the transformants. In contrast, control strains pR3 and pR6 show the presence of a 1200 bp amplicon, confirming the absence of *azpA* insert in the cassette. *P. verrucosus* FAE27 (WT) was included as negative control. Lane M corresponds to the standard O’GeneRuler 1 kb DNA Ladder (Thermo Fischer Scientific, Braunschweig, Germany), with relevant sizes indicated on the left. (**C**) qRT-PCR analysis of the expression of *azpA* in the strains. Wild-type *P. verrucosus* FAE27 (WT) and transformants pR3 and pR6 containing the empty pJLH-RNAi vector serve as controls. Error bars represent the standard deviation of three replicates in three different experiments. Statistically significant differences were denoted at *p* < 0.05 (***) using one-way ANOVA and Tukey’s test. (**D**) Metabolic profiles at 530 nm of *P. verrucosus* FAE27 (WT) and transformants obtained through RNAi-mediated gene silencing. Alongside WT, strains azpA-pR1 (representative of white transformants), azpA-pR13 (representative of red transformants), and pR6 (control transformed with empty pJLH-RNAi) are included. Note the absence of absorbance at 530 nm in the white transformant azpA-pR1 in comparison to WT and red transformants.

**Figure 3 jof-10-00157-f003:**
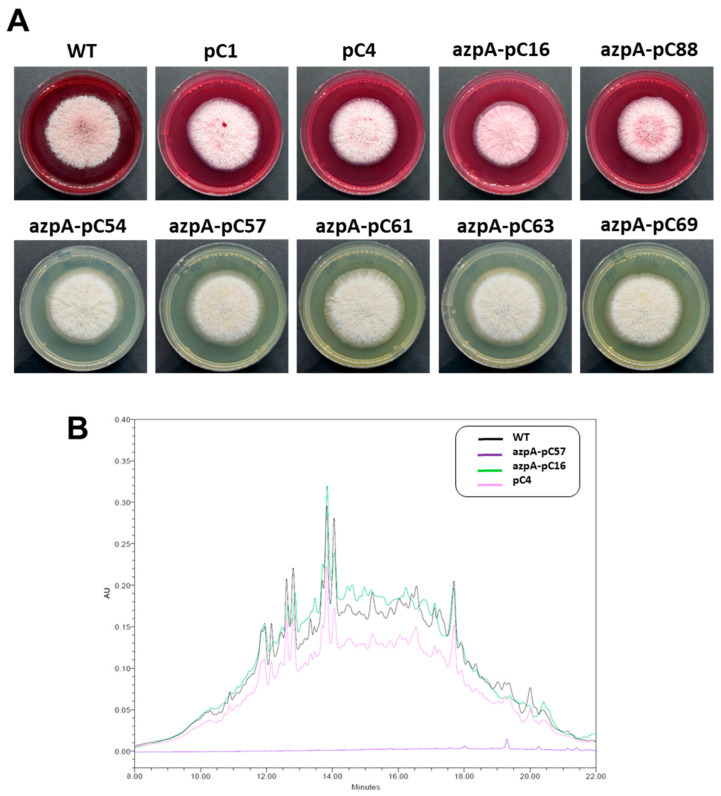
Phenotypic variations in *Pseudogymnoascus verrucosus* FAE27 and representative strains obtained from pFC332-azpA and pFC332 transformations. (**A**) *P. verrucosus* FAE27 (WT) exhibits its characteristic red pigmentation. Following transformation with pFC332-azpA, the phenotypes of the obtained transformants vary. Most transformants (azpA-pC54, azpA-pC57, azpA-pC61, azpA-pC63, and azpA-pC69) display a white phenotype, while some maintain the red coloration, resembling the wild-type appearance (azpA-pC16 and azpA-pC88). Additionally, two control transformants (pC1 and pC4), harboring the empty pFC332 plasmid without the CRISPR-Cas9 expression cassette, were included. As expected, *P. verrucosus* FAE27 and transformants pC1 and pC4 have the same phenotype. (**B**) Metabolic profiles at 530 nm of *P. verrucosus* FAE27 (WT) and transformants obtained by CRISPR/Cas9 methodology: azpA-pC57 (representative of white transformants) and azpA-pC16 (representative of red transformants). Additionally, the control strain pC4, transformed with the empty pFC332 plasmid, is included. Note the absence of absorbance at 530 nm in the white transformant azpA-pC57 in comparison to WT and red transformants.

**Figure 4 jof-10-00157-f004:**
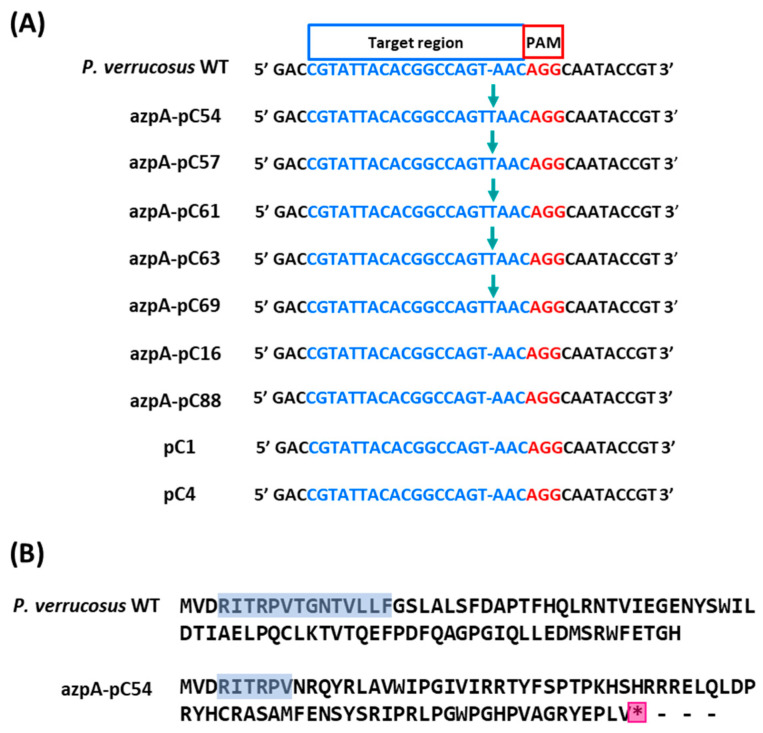
Comparison of the *azpA* gene target region in *P. verrucosus* FAE27 and selected transformants. (**A**) Sequence alignment of the target region of the *azpA* gene (highlighted in blue) in *P. verrucosus* FAE27 (labeled as *P. verrucosus* WT), white transformants (azpA-pC54, azpA-pC57, azpA-pC61, azpA-pC63, and azpA-pC69), and red transformants (azpA-pC16 and azpA-pC88). Control transformants pC1 and pC4 are also included. The protospacer adjacent motif (PAM) sequences are highlighted in red. All white transformants exhibit a consistent single nucleotide insertion at the same position (indicated by arrows). (**B**) Analysis of the amino acid sequence of AzpA protein from *P. verrucosus* FAE27 and the representative transformant azpA-pC54. The amino acid sequence deduced from the *azpA* target region is highlighted in blue. In strain azpA-pC54, this region displays frameshifts and a premature stop codon (highlighted with a pink asterisk). The same alterations were observed in all other white transformants. Conversely, all red transformants exhibit the same intact sequence as *P. verrucosus* WT.

**Table 1 jof-10-00157-t001:** Analysis of the deduced proteins encoded by the *azp* genes from *Pseudogymnoascus verrucosus* FAE27 and their putative functions.

			Closest Characterized Homologues		
Protein Name	Size (Aminoacids)	Putative Function	Protein Name (Organism)	GenBank Accession Number	Identity (%)
AzpA	2698	Non-reducing polyketide synthase	Conidial yellow pigment biosynthesis polyketide synthase (*Monascus pilosus*)	AGN71604	61
AzpB	2315	Highly reducing polyketide synthase	Polyketide synthase (*Aspergillus niger*)	EHA28244	45
AzpC	374	Ketoreductase	Aldehyde reductase (*Monascus pilosus*)	AGN71608	50
AzpD	455	O-acetyltransferase	Acetyltransferase (*Monascus pilosus*)	AGN71607	43
AzpE	445	FAD monooxygenase	Monooxygenase (*Phoma* sp.)	QCO93109	53
AzpF	268	Serine hydrolase	Amino oxidase/esterase (*Monascus pilosus*)	AGN71609	58
AzpG	364	Enoyl reductase	Putative quinone-oxidoreductase-like protein (*Monascus pilosus*)	AGN71610	50
AzpH	644	FAD oxidase	Isoamyl alcohol oxidase (*Penicillium expansum*)	AIG62142	36
AzpI	368	Cytochrome P450	BuaG cytochrome P450 (*Aspergillus burnettii*)	QBE85647	49
AzpJ	482	FAD oxidase	FAD oxidase (*Aspergillus niger*)	EHA28243	47
AzpK	216	Transporter	AflT transporter (*Aspergillus flavus*)	AAS90069	45
AzpL	119	Transcription factor	Putative citrinin biosynthesis transcriptional activator CtnR (*Monascus pilosus*)	AGN71605	51

## Data Availability

Data are contained within the article and Appendix A.

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
