# Peer review of "Expanding the Toolbox for Genetic Manipulation in Pseudogymnoascus: RNAi-Mediated Silencing and CRISPR/Cas9-Mediated Disruption of a Polyketide Synthase Gene Involved in Red Pigment Production in P. verrucosus"

_jof, 2024, doi:10.3390/jof10020157_

Round 1

Reviewer 1 Report

Comments and Suggestions for Authors

Fungi within the Pseudogymnoascus genus, notably P. destructans causing severe bat disease, have gained attention. Despite increased genome sequencing, the genetic landscape remains unexplored due to limited molecular tools. This study, utilizing an Antarctic P. verrucosus strain, achieved successful RNAi-mediated gene silencing and CRISPR/Cas9 disruption, targeting the red pigment biosynthesis gene, azpA. Silencing resulted in a 90% reduction, yielding white transformants, while CRISPR/Cas9 induced a 73% inactivation, also producing white phenotypes. This marks a significant breakthrough, enabling comprehensive genetic investigations in this underexplored fungal genus.

1.    In the passage spanning lines 285-287, the authors assert that ‘As described in the literature concerning azaphilone BGCs, the NR-PKS enzyme plays a pivotal role in the formation of the core structure of azaphilone [44].’  However, my review of the pertinent literature has not yielded corroboration of this specific information. Could you kindly provide guidance on the precise source or elucidate further on this assertion for clarification?

2.    The authors are encouraged to enhance the clarity of their manuscript by including supplementary material that visually depicts the Vector cassette. A graphical representation of the Vector cassette for both RNAi and CRISPR-Cas9 in the supplementary section would be particularly beneficial for readers, facilitating a more comprehensive understanding of the experimental setup and methodology employed in this study.

3.    Have the authors undertaken subculturing of the transformed colonies? Additionally, could the authors provide insights into whether the morphological characteristics of these colonies reverted to those resembling the wild type (WT) during subsequent subcultures?

4.    It seems like CRISPR-Cas9 is mor effective than RNAi, can you discuss more on this point?

Author Response

Fungi within the Pseudogymnoascus genus, notably P. destructans causing severe bat disease, have gained attention. Despite increased genome sequencing, the genetic landscape remains unexplored due to limited molecular tools. This study, utilizing an Antarctic P. verrucosus strain, achieved successful RNAi-mediated gene silencing and CRISPR/Cas9 disruption, targeting the red pigment biosynthesis gene, azpA. Silencing resulted in a 90% reduction, yielding white transformants, while CRISPR/Cas9 induced a 73% inactivation, also producing white phenotypes. This marks a significant breakthrough, enabling comprehensive genetic investigations in this underexplored fungal genus.

R: Dear reviewer, we sincerely appreciate your positive feedback and valuable constructive criticism regarding our work. In the following paragraphs, we will address all your specific concerns.

  1. In the passage spanning lines 285-287, the authors assert that ‘As described in the literature concerning azaphilone BGCs, the NR-PKS enzyme plays a pivotal role in the formation of the core structure of azaphilone [44]. However, my review of the pertinent literature has not yielded corroboration of this specific information. Could you kindly provide guidance on the precise source or elucidate further on this assertion for clarification?

R: Thank you for bringing this to our attention. Certainly, we can provide more clarity on this assertion. To date, five main known biosynthetic pathways have been described for azaphilone biosynthesis: the Monascus azaphilone pathway, the Aspergillus azaphilone pathway, the citrinin-type azaphilone pathway (found in several fungi), the Chaetomium azaphilone pathway, and the Hypoxylon azaphilone pathway. These pathways are described in the review that we referenced in our manuscript (reference [44]) as the source for the assertion on lines 285-287. This is the most recent review on the topic. In all the five pathways mentioned above, the first step in the biosynthesis of azaphilones is the production of an orcinaldehyde intermediate. In four of the pathways, this intermediate is produced by the joint action of an NR-PKS and an HR-PKS (see section 2.2 in reference [44]), while in the Monascus azaphilone pathway, only the NR-PKS enzyme is required for the formation of the orcinaldehyde intermediate (see Figure 3 and section 2.1 in reference [44]). In summary, the NR-PKS, alone, or along with an HR-PKS, plays a fundamental role in the synthesis of the core of azaphilones (the orcinaldehyde intermediate). We agree with the reviewer that our sentence is quite general and does not address this information properly, so we have modified the sentence for clarification.

  1. The authors are encouraged to enhance the clarity of their manuscript by including supplementary material that visually depicts the Vector cassette. A graphical representation of the Vector cassette for both RNAi and CRISPR-Cas9 in the supplementary section would be particularly beneficial for readers, facilitating a more comprehensive understanding of the experimental setup and methodology employed in this study.

R: Thank you for bringing this to our attention. Following your feedback, we have included a graphical representation of both vectors. In addition, we have included the steps necessary for their construction to enhance clarity.

  1. Have the authors undertaken subculturing of the transformed colonies? Additionally, could the authors provide insights into whether the morphological characteristics of these colonies reverted to those resembling the wild type (WT) during subsequent subcultures?

R: Yes, we have conducted subculturing of the strains. They were subcultured between 6 to 12 times, depending on the specific strain. As of now, there has been no discernible reversion of the phenotype in any of the strains.

  1. It seems like CRISPR-Cas9 is mor effective than RNAi, can you discuss more on this point?

R: We appreciate the reviewer's interesting suggestion. Accordingly, we have added a paragraph in the Discussion section addressing this point, including two additional references.

Reviewer 2 Report

Comments and Suggestions for Authors

In this study, RNAi-mediated gene silencing and CRISPR/Cas9-mediated disruption were successfully implemented in P. verrucosus. Silencing or inactivating of the azpA gene involved in red pigment biosynthesis resulted in a white phenotype. These methods lays the foundation of functional genetics research of P. verrucosus. However, as the research on methodology, the authors conducted the research on a single gene only, and designed only one RNAi fragment or sgRNA, lacking analysis of some key factors for RNAi silencing or CRISPR/Cas9-mediated inactivation. For example, RNAi fragment length, different target sequences on CRISPR-Cas9 inactivation, or transformation efficiency of the recombinant plasmids, etc. The article only conveyed to readers these two methods can be used in P. verrucosus, without providing suggestions on how to use, which resulted in poor readability, novelty, and scientific rationality of the article. Thus, the author should supplement at least the experiments above to enrich the content of the article

1.     Figure 1b has no effect on supporting the content of the paper and can be removed.

2.     To make it easier to understand for readers, authors should display the recombinant plasmid map, and provide a schematic diagram of the plasmid construction process.

3.     The author only indicated the number of transformants obtained (line 305 and 344), but did not present the number of cells used for transformation or the transformation efficiencies. As it is one of the key factors that gene manipulation techniques focus on, the authors should present these data,

4.     Figure 5 showed the phenotypic results of the strains. The author can integrate into Figure 2 and 3, respectively.

Author Response

In this study, RNAi-mediated gene silencing and CRISPR/Cas9-mediated disruption were successfully implemented in P. verrucosus. Silencing or inactivating of the azpA gene involved in red pigment biosynthesis resulted in a white phenotype. These methods lays the foundation of functional genetics research of P. verrucosus. However, as the research on methodology, the authors conducted the research on a single gene only, and designed only one RNAi fragment or sgRNA, lacking analysis of some key factors for RNAi silencing or CRISPR/Cas9-mediated inactivation. For example, RNAi fragment length, different target sequences on CRISPR-Cas9 inactivation, or transformation efficiency of the recombinant plasmids, etc. The article only conveyed to readers these two methods can be used in P. verrucosus, without providing suggestions on how to use, which resulted in poor readability, novelty, and scientific rationality of the article. Thus, the author should supplement at least the experiments above to enrich the content of the article

R: Thank you for taking the time to review our manuscript and providing valuable feedback. We appreciate your careful consideration of our work. Next, we will address your specific concerns.

In relation to additional experiments to address the parameters of RNAi fragment length, it is noteworthy that the plasmid pJL43-RNAi has demonstrated successful utilization across various fungal species since 2008. These include Penicillium chrysogenum, Acremonium chrysogenum, Fusarium oxysporum, Aspergillus niger, Trichoderma longibrachiatum, and Penicillium roqueforti. According to the existing literature, pJL43-RNAi has been efficiently employed to downregulate at least 50 different genes in these fungi over the years. Consequently, a substantial amount of information has been accumulated concerning the use of varying fragment lengths with this plasmid over time. Attached to this response, you will find a table detailing the fragment lengths utilized for each of the 50 genes, accompanied by the respective bibliographic citations. The table demonstrates that, since 2008, 49 different insert sizes have been tested in pJL43-RNAi, all yielding successful results. The sizes of these fragments span a broad range, from 257 to 1,055 base pairs, with the most frequently observed sizes falling within the range of 303-479 base pairs. In the light of this extensive experience, it becomes clear that fragment sizes around 300-500 base pairs perform effectively in this plasmid. In line with this accumulated evidence, we selected a fragment size of 413 base pairs for our experiments, resulting in successful outcomes. In summary, the 50 RNAi experiments conducted and published since 2008 provide robust evidence concerning the suitable range of RNAi fragment lengths for the use of pJL43-RNAi in fungi. Based on this comprehensive body of evidence, we have determined that conducting the suggested experiment is unnecessary and would not significantly contribute to the specific objectives of our research, nor would it enhance the existing knowledge about pJL43-RNAi.

In the context of CRISPR-Cas9 experiments, we meticulously selected the optimal sgRNA, adhering to established best practices. The targeted gene, azapA, exhibits substantial genomic length, encompassing 8,151 nucleotides, and encodes a polyketide synthase (PKS) featuring multiple catalytic domains. By this reason and adhering to literature recommendations for enhanced CRISPR efficiencies (as exemplified in Nucleic Acids Res. 2022 50: 3616–3637), we decided to position the target sequence within the first exon, proximal to the N-terminus of the protein. This strategic placement ensures the induction of frame shifts and multiple stop codons from the gene's outset, producing a comprehensive protein structure disruption, including all domains. Consequently, for sgRNA design, a region spanning the initial 180 base pairs of the N-terminal encoding region of AzpA was employed as input in the sgRNACas9 v3.0.5 program. Thirteen putative sgRNAs were generated by the program, with five being promptly discarded due to the elevated risk of off-target effects arising from high similarity with other genes in the Pseudogymnascus genome. The remaining eight sgRNAs underwent meticulous scrutiny within the region spanning 12 nucleotides adjacent to the Protospacer Adjacent Motif (PAM). Empirical evidence dictates that sequences that match PAM and the 12 PAM proximal nucleotides can also produce off-target effects (Nat Biotechnol. 2013; 31: 827–32; Nucleic Acids Res. 2013; 4; PLoS ONE 10(7): e0133085). Following this analysis, one additional sgRNA was excluded. The remaining seven sgRNAs underwent an additional efficiency test due to the use of the proline tRNA promoter, activated by RNA polymerase III. It has been documented that sgRNAs expressed by RNA polymerase III exhibit diminished efficiency when specific motifs are present at the 3´end of the sgRNA (motifs described in Cell Rep. 2019 Jan 29; 26(5): 1098–1103.e3.). After this evaluation, three additional sgRNAs were discarded due to the potential risk of low efficiency. The outcome of this meticulous analysis revealed that 4 sgRNAs were potentially equivalent for CRISPR-Cas9 experiments. However, in alignment with the criterion of proximity to the region encoding the N-terminus of the protein, as mentioned previously, the sequence 5´CGTATTACACGGCCAGTAAC 3´ was selected, spanning nucleotides 10-23 of the gene. This sequence was the closest to the translation start region, ensuring the induction of frame shifts and multiple stop codons from the gene's outset.

The results obtained in our experiments support the correct choice of the sgRNA. As detailed in the manuscript, we achieved a 76% rate of mutants displaying a white phenotype, falling within the high-efficiency range for gene knockout in filamentous fungi (Appl Microbiol Biotechnol. 2017, 101:7435-7443; Gene 2017, 627:212-221). At this point, it is important to underscore two key aspects: firstly, the comprehensive sequencing of the target region in all mutants, including those maintaining the red phenotype, substantiates results not solely through phenotypic observation but also through genetic confirmation. Secondly, results were further validated through an alternative technique (RNAi-mediated gene silencing). Thus, we employed two distinct methodologies confirming the causal role of the azpA gene in the red coloration of Pseudogymnauscus verrucosus. In summary, after careful consideration of your comments, along with a deep analysis of the methodology for sgRNA obtainment, the achieved high targeting efficiency, the pertinent previous literature, and results corroboration by an alternative technique (RNAi-mediated gene silencing), we consider that additional experiments using the other three sgRNAs will yield comparable results to those obtained with the utilized sgRNA. Such additional experiments would not significantly contribute to the scientific understanding of the specific gene under investigation and would only unnecessarily delay the publication of the manuscript.

Regarding the transformation efficiency of the recombinant plasmids, these data have been incorporated into the revised version of the manuscript. It is important to underscore that the methodology employed for the transformation of Pseudogymnoascus verrucosus was previously elucidated in a 2019 publication (Front. Microbiol. 2019, 10: 2675). In that comprehensive study, Diaz et al. meticulously detailed the experimental protocol, encompassing all pertinent variables inherent to such transformations. It is noteworthy that the strain utilized in the current manuscript is the same employed in the aforementioned work by Diaz et al. Therefore, all aspects pertaining to the optimization of P. verrucosus FAE27 transformation were exhaustively addressed in that prior publication, rendering unnecessary the repetition of such optimizations in the present manuscript.

Finally, regarding your concern that the article conveyed only that these two methods can be used on P. verrucosus without providing suggestions on how to use them, we appreciate your attention to detail. In response to your comments, we have incorporated a comprehensive paragraph into the discussion section to explicitly address this issue. This newly added paragraph delves into various scenarios for applying both RNAi and CRISPR/Cas9 techniques in Pseudogymnoascus, providing specific examples of how these methods could be employed. Furthermore, we offer guidance on when one technique might be more advantageous or suitable compared to the other, enhancing the practical understanding of their applicability in the context of Pseudogymnoascus. We believe that this extended discussion not only addresses your concern but also significantly enriches the article by providing valuable insights into the practical implementation of RNAi and CRISPR/Cas9 in Pseudogymnoascus.

  1. Figure 1b has no effect on supporting the content of the paper and can be removed.

R: We appreciate your suggestion. Accordingly, we have removed Figure 1b.

  1. To make it easier to understand for readers, authors should display the recombinant plasmid map, and provide a schematic diagram of the plasmid construction process.

R: Thank you for bringing this to our attention. Following your feedback, we have included a Supplementary figure displaying graphical representation of plasmids. In addition, we have included schematic diagram of plasmid construction processes.

  1. The author only indicated the number of transformants obtained (line 305 and 344), but did not present the number of cells used for transformation or the transformation efficiencies. As it is one of the key factors that gene manipulation techniques focus on, the authors should present these data,

R: Thank you for bringing this to our attention. The number of cells and quantity of plasmid used in each experiment were included in the respective Methods section. Transformation efficiencies were included in Results section.

  1. Figure 5 showed the phenotypic results of the strains. The author can integrate into Figure 2 and 3, respectively.

R: We appreciate the interesting reviewer's suggestion. We have integrated Fig. 5 into Fig. 2 and 3. Accordingly, we also have adjusted Figure captions and the main text.

Reviewer 3 Report

Comments and Suggestions for Authors

Dear authors,

I am writing to express my sincere appreciation for your paper entitled "Expanding the toolbox for genetic manipulation in Pseudogymnoascus: RNAi-mediated silencing and CRISPR/Cas9-mediated disruption of a polyketide synthase gene involved in red pigment production in P. verrucosus"

The genetic landscape of Pseudogymnoascus has remained largely unexplored, primarily due to the limited availability of suitable molecular tools for genetic manipulation. This study addresses this gap by successfully implementing RNAi-mediated gene silencing and CRISPR/Cas9-mediated disruption in Pseudogymnoascus, using an Antarctic strain of Pseudogymnoascus verrucosus as a model. Your study successful application of RNAi-mediated gene silencing and CRISPR/Cas9-mediated disruption marks a significant advancement in Pseudogymnoascus research. It not only provides valuable insights into the functional genetics of this underexplored fungal genus but also opens new avenues for comprehensive investigations in this field.

Top of Form

I found that this research article is well-structured, with clear objectives and a logical flow of information and results. The authors have demonstrated a thorough understanding of the subject matter and have meticulously used a wide range of literature sources.

However, there is a need for an improvement in some parts of the manuscript.

These include:

Title:

Pag 1, Line 5: Could you please delete point (.) at the end of your MS title?

1. Introduction:

Page 2, Line 47: Would you kindly include the Mycobank link in the Reference list? This request is based on your mention that the examined genera contain 28 species according to Mycobank.

Page 2, Line 53: Could you mind substituting the term "compounds" for "molecules"?

Page 2, Lines 53: Could you paraphrase this part of the sentence: “produce molecules with interesting chemical structures, encompassing” as for example “produce different classes of compounds such as”

2. Materials and methods

Well described.

3. Results and discussion

Good Job, indeed.

Top of Form

In conclusion, I recommend a minor revision of your manuscript.

Author Response

Dear authors,

I am writing to express my sincere appreciation for your paper entitled "Expanding the toolbox for genetic manipulation in Pseudogymnoascus: RNAi-mediated silencing and CRISPR/Cas9-mediated disruption of a polyketide synthase gene involved in red pigment production in P. verrucosus"

The genetic landscape of Pseudogymnoascus has remained largely unexplored, primarily due to the limited availability of suitable molecular tools for genetic manipulation. This study addresses this gap by successfully implementing RNAi-mediated gene silencing and CRISPR/Cas9-mediated disruption in Pseudogymnoascus, using an Antarctic strain of Pseudogymnoascus verrucosus as a model. Your study successful application of RNAi-mediated gene silencing and CRISPR/Cas9-mediated disruption marks a significant advancement in Pseudogymnoascus research. It not only provides valuable insights into the functional genetics of this underexplored fungal genus but also opens new avenues for comprehensive investigations in this field.

I found that this research article is well-structured, with clear objectives and a logical flow of information and results. The authors have demonstrated a thorough understanding of the subject matter and have meticulously used a wide range of literature sources.

R: Dear reviewer, we sincerely appreciate your insightful feedback and positive evaluation of our work.

However, there is a need for an improvement in some parts of the manuscript.

These include:

Title:

Pag 1, Line 5: Could you please delete point (.) at the end of your MS title?

R: Thank you for drawing our attention on this error. It was corrected.

  1. Introduction:

Page 2, Line 47: Would you kindly include the Mycobank link in the Reference list? This request is based on your mention that the examined genera contain 28 species according to Mycobank.

R: Thank you for your valuable input. Based on your suggestion, we have incorporated a new Table in the Supplementary material, detailing the names of the species and their MycoBank identifier. On the MycoBank page, you can use the identifier number to obtain all the information about these species. Please note that since the original submission date of this manuscript (November 2023, 3rd), three new species of Pseudogymnoascus have been added to MycoBank. Therefore, in this revised version, we have updated the number of species (from 28 to 31 species).

Page 2, Line 53: Could you mind substituting the term "compounds" for "molecules"

R: We have made the requested change in the manuscript.

Page 2, Lines 53: Could you paraphrase this part of the sentence: “produce molecules with interesting chemical structures, encompassing” as for example “produce different classes of compounds such as”

R: We have made the requested change in the manuscript.